# Carbon and Nitrogen Availability Drives Seasonal Variation in Soil Microbial Communities along an Elevation Gradient

**Xiaoling Xiong** [1,2], **Maokui Lyu** [1,2,\*], **Cui Deng** [1,2], **Xiaojie Li** [1,2], **Yuming Lu** [1,2], **Weisheng Lin** [1,2], **Yongmeng Jiang** [1,2] and **Jinsheng Xie** [1,2,\*]

[1] Key Laboratory for Humid Subtropical Eco-Geographical Processes of the Ministry of Education, School of Geographical Sciences, Fujian Normal University, Fuzhou 350007, China
[2] Sanming Forest Ecosystem National Observation and Research Station, Sanming 365002, China
\* Correspondence: 228lmk@163.com (M.L.); jshxie@163.com (J.X.)

**Abstract:** Changes in soil abiotic and biotic properties can be powerful drivers of feedback between plants and soil microbial communities. However, the specific mechanisms by which seasonal changes in environmental factors shape soil microbial communities are not well understood. Here, we collected soil samples from three sites along an elevational gradient (200–1200 m) in subtropical forests with unvarying canopy vegetation. We used an elevation gradient with similar annual precipitation but a clear temperature gradient, and phospholipid fatty acids (PLFAs) were measured to determine the seasonal variations in the composition of soil microbial communities in response to rising temperatures. Our results showed that the abundance of Gram-negative bacteria and total PLFAs were the lowest at low elevations in winter, and the ratio of Gram-positive to Gram-negative bacteria decreased with increasing elevation. However, the biomass of other microbial groups was the highest at medium elevations in summer, with the exception of actinomycetes species and fungi. Regardless of seasonal changes, soil fungal biomass tended to increase with increasing elevation. Moreover, in summer, microbial carbon use efficiency (CUE) increased with increasing elevation, whereas an opposite trend was observed in winter. Redundancy analysis and structural equation modeling showed that the dissolved organic carbon in soil was the main factor affecting the microbial communities along the elevation gradient in winter, whereas in summer, the microbial community structure was driven by shifting nitrogen availability, with both being associated with changing microbial CUE. As such, this study demonstrates distinct seasonal changes in the soil microbial community composition across an elevation gradient that are driven by carbon and nitrogen resource availability and shifts in microbial CUE. Furthermore, our results suggest that the interaction of underground plant roots and microbes drives changes in resource availability, thereby resulting in seasonal variation in soil microbial community composition across an elevation gradient.

**Keywords:** *Cunninghamia lanceolata* (Lamb.) Hook.; forests; elevation gradient; microbial community; microbial C use efficiency; plant–microbial interactions

## 1. Introduction

Soil microorganisms are a vital component of forest ecosystems due to their role in driving and regulating material circulation and biochemical processes [1–3]. These communities regulate ecosystem balance through basic ecological processes such as mineralization and decomposition [4]. Previous studies have shown that soil microorganisms are an essential medium connecting surface and soil ecologies, thereby promoting the growth of plants and maintaining nutrient circulation [5]. These studies suggest that the soil microbial community is a key determinant of function and that direct changes to the structure of this community may alter not only microbial function but nitrogen and carbon soil dynamics as well [6].

It has been extensively shown that microbial biomass as well as community composition are highly interrelated with the seasons [7,8]. The changing of seasons leads to changes in soil temperature, water, and other environmental factors, which may be the key factor driving the seasonal dynamics of forest soil microbial communities [9,10]. This, in turn, impacts the decomposition rate of soil organic matter and nutrient cycling [11]. In addition, vegetation phenology and vegetation cover vary significantly with the seasons, and their contributions to soil organic matter and nitrogen content may influence the composition of the microbial community [12]. Plants excrete active carbon through roots and input substrate through litter, which alters the effectiveness of soil microorganisms on carbon and nitrogen processing [13].

Elevation gradients have been recognized as useful "natural experiments" for climate change studies, as dramatic changes in climatic and biological characteristics are observed over short geographic distances [14–17]. Elevation gradient studies help researchers understand the influence of environmental factors such as temperature, soil moisture, and nutrient availability on soil microbial communities in a way not achievable through conventional manipulation experiments [18–20]. However, previous studies on microbial properties along elevation gradients have shown mixed results. For example, researchers have reported that microbial communities show variable patterns of abundance and structure with increasing elevation in different ecosystems. These patterns have included a continuous increase [21], a continuous decrease [22,23], a humpback pattern [24], and no consistent pattern [25]. A global meta-analysis study performed in 2017 also showed no consensus about the patterns of microbial communities or activities along elevation gradients [26]. The conflicting results of these studies indicate that more work is needed to disentangle the effects of complex environmental conditions along elevation gradients on soil microbial communities.

Despite its relevance, knowledge about the seasonal variation in microbial communities along elevation gradients is still scarce and contains critical knowledge gaps. Over the annual cycle, soil microorganisms face large seasonal variations in environmental conditions, such as temperature, moisture, and resource availability [27]. Further investigation into the response of soil microbial communities to changes in elevation during different seasons can improve our understanding of the impact of climate change on soil, plants, and microorganisms. However, the drivers of seasonal variation in soil microbial communities over an elevation gradient are unclear. Climatic and edaphic variables (e.g., pH and C/N) are frequently reported as key factors shaping microbial elevational patterns [18], but these variables have limited explanatory power. For example, plant and microbial interaction-induced changes in soil microbial community composition are often ignored [28].

To address this issue, in this study, we considered the elevation gradient as the carrier of climate change and *Cunninghamia lanceolata* forests at different altitudes in Wuyishan National Park in winter and summer as the research subjects. The Wuyishan National Park is one of the best-preserved virgin forest vegetation reserves in southeast China, where species resources are rich and diverse. It is therefore an ideal experimental site for us to discuss the influence of seasons on the elevation gradient of the soil microbial community structure in forests. For example, in summer, plants need more nutrients to supply their growth requirements, so there will be intense nutrient competition between plants and microorganisms. In winter, plants grow slowly and do not need as many nutrients, while microbes need more carbon to adapt to lower temperatures. We hypothesized that across the elevation gradient, seasonal resource availability for heterotrophic microbial communities in soil will vary with increasing elevation and will thus modify the activity, biomass, and composition of those communities.

## 2. Materials and Methods

### 2.1. Site Description

The experimental sites for this study are located in Wuyishan National Park, Fujian Province, China (27°33′–27°54′ N, 117°27′–117°51′ E). The park was established in 1979 and covers an area of about 570 km². Wuyi Mountain is the main peak, with an elevation of 2157.8 m, and the site is exposed to a subtropical monsoonal climate. Our study was conducted in *Cunninghamia lanceolata* forests at different elevations spanning ~1000 m (200 m to 1200 m) at low (200 m), medium (700 m), and high (1200 m) elevations. The mean annual temperature ranges from 20.2 °C to 14.2 °C (from low to high elevations). *Cunninghamia lanceolata* is the main dominant species, with few other species present. Understory shrubs mainly consist of *Eurya japonica Thunb*, *Ardisia japonica*, *Tectaria phaeocaulis*, *Microlepia Presl*, and other ferns. Table 1 lists the basic vegetation information of the three sites.

**Table 1.** Site characteristics across the studied elevations in *Cunninghamia lanceolata* forests. Values are means ± standard deviation (*n* = 4). Different lowercase letters indicate significant differences between elevations.

| Elevation (m a.s.l) | 200 | 700 | 1200 |
|---|---|---|---|
| Stand density (Tree hm$^{-2}$) | 1696 | 1384 | 542 |
| Tree height (m) | 20.2 | 25.2 | 25.0 |
| Diameter at breast height (cm) | 16.7 | 16.4 | 28.8 |
| Annual temperature (°C) | 20.2 | 17.6 | 14.2 |
| Annual precipitation (mm) | 2411 | 2374 | 2481 |
| Soil pH | 5.05 (0.32) a | 4.86 (0.21) a | 4.23 (0.02) b |
| Bulk density (g cm$^{-3}$) | 1.10 (0.07) a | 0.74 (0.11) b | 0.78 (0.08) b |
| Soil organic carbon (g kg−1) | 26.14 (3.24) b | 50.53 (6.50) a | 53.31 (7.21) a |
| Total nitrogen (g kg−1) | 1.90 (0.29) b | 3.15 (0.34) a | 2.62 (0.38) a |
| Soil C/N | 13.76 (0.82) c | 16.08 (0.39) b | 20.35 (1.51) a |
| Total phosphorus (g kg−1) | 0.39 (0.02) a | 0.32 (0.02) b | 0.28 (0.01) c |
| Soil temperature (°C) | 18.85 (0.08) a | 16.74 (0.07) b | 14.65 (0.12) c |
| Soil moisture (%) | 10.18 (1.79) b | 14.47 (0.97) a | 15.00 (0.54) a |

The soil types of the three elevations were classified as red soil (200 m), yellow red soil (700 m), and yellow soil (1200 m) according to the Chinese Soil Classification System. These are equivalent to Ultisols and Inceptisols in the United States Department of Agriculture (USDA) Soil Taxonomy Classification System [28].

### 2.2. Sample Collection and Preparation

Field sampling was performed in January 2018 (winter) and in June 2018 (summer) at three elevations of *Cunninghamia lanceolata* forests. At each elevation, four 4 m × 4 m subplots were set up, and each of the four plots was separated by at least five meters. For statistical purposes, we treated each of the four plots as independent replicates (*n* = 4). In each subplot, ten replicate soil cores from the topsoil (0–10 cm depth) were collected randomly using a hand auger (diameter 3.5 cm), and the 10 cm depth was entirely within the A horizon at each site. Soil cores were pooled into one composite sample from each quadrant. All soil samples were packed and immediately brought back to the laboratory for processing. Tweezers were used to pick out visible root and plant debris, and the soil was sieved through a 2 mm mesh. Subsequently, we divided the soil into two parts: one part was stored at 4 °C for the determination of soil-available nutrients and the composition of microbial community, and the remaining soil was air-dried and stored in airtight plastic bags until analysis within one month. From January to December of 2018, we used a hand-held long-rod electron temperature probe (SK-250WP, Sato Keiryoki, Kanda, Japan) and a Time Domain Reflectometer (TDR300, Spectrum, Aurora, CO, USA) to monitor soil temperature (Ts) and soil moisture (SM) (Figure A1).

### 2.3. Soil Chemical Analysis

Soil pH was measured with a pH meter on a paste of 1:2.5 (w:v) air-dried soils and deionized water. Soil organic carbon (SOC) and total nitrogen (TN) contents were determined from the finely ground (<0.15 mm) subsamples of air-dried soil using a Vario MAX CN elemental analyzer (Vario Max CN, Elementar, Landgenselbold, Germany). Soil total phosphorus (TP) was analyzed by the $HClO_4$-$H_2SO_4$ acid digestion method in a continuous flow analyzer (San++, SKALAR Corporation production, Breda, The Netherlands) [29]. Soil dissolved organic carbon (DOC) and dissolved organic nitrogen (DON) were extracted from 10g of field-fresh soil in 40 mL deionized water [30]. The DOC and DON concentrations were determined using a TOC analyzer (TOC-VCPH, Shimadzu, Kyoto, Japan) and a continuous flow analyzer, respectively. The concentrations of mineral nitrogen (MN) were based on the sum of $NH_4^+$ and $NO_3^-$; 5 g of freshly collected soil was extracted with a 2 mol $L^{-1}$ KCl solution. Available phosphorus (AP) was extracted with Mehlich III extract and then measured with a continuous flow analyzer [31].

### 2.4. Biomass and Composition of Microbial Communities

Soil microbial biomass carbon (MBC) and nitrogen (MBN) were determined using the chloroform ($CHCl_3$) fumigation and potassium sulfate ($K_2SO_4$) extraction techniques described by [32,33], and the TOC analyzer (TOC-VCPH, Shimadzu, Kyoto, Japan) was then used to determine MBC, and a continuous flow analyzer (San++, SKALAR Corporation production, Breda, The Netherlands) was used to determine MBN. The universal conversion factors of MBC and MBN were 0.45 and 0.54, respectively.

The microbial community structure was determined using phospholipid fatty acid (PLFA) analysis as described by White et al. [34] and Bardgett et al. [35]. In summary, we used a mixture of chloroform–methanol–phosphate buffer (1:2:0.8 *v/v/v*) to extract lipids from 3 g freeze-dried soil, and the extraction process had two stages: the chloroform stage and citric acid buffer stage. The extracted phospholipids were methylated to form fatty acid methyl esters (FAMEs), which were identified on a gas chromatograph (GC) (Agilent 6890 N, Santa Clara, CA, USA) equipped with the MIDI Sherlock Microbial Identification System. The concentration of each PLFA was calculated on the basis of the 19:0 internal standard concentrations. Relative nanomole per gram of dry soil was used to express the abundance of individual fatty acids using standard nomenclature [36]. The PLFA biomarkers i14:0, i15:0, a-15:0, i16:0, i17:0, and a17:0 represented Gram-positive bacteria (GP) [37], and 16:1 w9c, 16:1 w7c, 18:1 w7c, 18:1 w5c, cy19:0 w7c, and cy17:0 w7c were biomarkers for Gram-negative bacteria (GN) [38,39]. Fungi were identified by the PLFAs 18:1w9c and 18:2 w6c [40], while 16:1 w5c was used to represent arbuscular mycorrhiza fungi (AMF) [41]. PLFAs 16:0 10-methyl, 17:0 10-methyl, and 18:0 10-methyl were used as markers for actinomycetes (ACT) [40,42]. The ratio of fungal to bacterial PLFAs (F:B) was used to estimate the relative importance of bacterial and fungal metabolism in the community.

### 2.5. Soil Enzymes Activities and Microbial Carbon Use Efficiency

We also studied soil enzymes that play key roles in the mineralization of C, N, and P in soil, including β-glucosidase (βG), cellulosebiohydrolase (CBH), nacetyl glucosaminidase (NAG), and acid phosphatase (AP-Tase), using moist soil stored at 4 °C. To obtain the specific enzyme activity index, we normalized the total enzyme activity using the total potential activity according to the MBC concentration. Based on Sinsabaugh et al. [43], we calculated the ratios of C, N, and P by determining the enzyme activity.

The microbial CUE was calculated based on the following equations for C:N stoichiometry:

$$CUE_{C:N} = CUE_{max} [S_{C:N}/(S_{C:N} + K_N)]$$
$$S_{C:N} = (1/EEA_{C:N}) (B_{C:N}/L_{C:N})$$

where $S_{C:N}$ is a scalar quantity to indicate the extent to which the distribution of enzyme activity offsets the difference between the DOC:TDN ratio of resources and microbial

biomass [44]. The half-saturation constant $K_N$ was set to 0.5. $CUE_{max}$ is the upper limit for microbial growth efficiency and was set to 0.6 according to the thermodynamic constraints [44]. $EEA_{C:N}$ is the ratio of the enzyme activity of C and N obtained directly from the environment, which is calculated as (BG + CBH)/NAG. $L_{C:N}$ is the ratio of labile organic matter.

*2.6. Statistical Analysis*

All statistical analyses were carried out using the IBM SPSS Statistics 21.0 software (SPSS Inc., Chicago, IL, USA), Origin 2021 (Origin Lab Corporation, Northampton, MA, USA), and CANOCO 5.0 software (Microcomputer Power Inc. Ithaca, NY, USA). One-way ANOVA comparisons with least significant difference (LSD) multiple comparisons were used to explore the differences in soil and microbial parameters (soil temperature, soil moisture, SOC, TN, TP, AP, MN, DOC, DON, microbial biomass, microbial PLFAs, CUE, and enzymes activities) at different elevations. An independent-sample t-test was used to determine the effects of season on soil properties and microbial parameters. The effects of elevation and season on microbial community structure, CUE, and enzyme activity were determined using two-way ANOVA followed by LSD post hoc multiple comparisons. The level of significance of the statistical tests was $\alpha = 0.05$. The heat maps for correlation analysis between the microbial community and soil characteristics were drawn in Origin 2021. Redundancy analysis (RDA) was used to test the elevation effects on soil microbial community structure and environmental variables. We used structural equation modeling (SEM) with AMOS 24.0 (AMOS Development Corporation, Chicago, IL, USA) to examine the key factors driving seasonal variations in microbial community structure, the prior models used for structural equation modeling analysis are presented in the Appendix A (Figure A2 and Table A2).

**3. Results**

*3.1. Soil Properties at Different Elevations*

The SOC and TN contents increased along the elevation gradient at a consistent rate, with the lowest levels at 200 m. However, soil TP content decreased linearly with elevation ($p < 0.05$, Table 1). The availability of soil C, N, and P were significantly different at different elevations. Soil DOC and DON concentrations increased with rising elevation both in winter and summer, and the difference in DOC was significant between elevations ($p < 0.05$, Figure 1). It was observed that the variation trend of soil $NH_4^+$ and $NO_3^-$ was dramatically different with increasing elevation. Soil $NO_3^-$ was the highest at 200 m and was lower at the other two elevations ($p < 0.05$, Figure 1). As an important part of mineral nitrogen, the variation trend of $NH_4^+$ is consistent with that of mineral nitrogen. In winter, the concentrations of soil AP at high elevations were significantly lower than that at 200 m and 700 m, whereas it was the highest at medium elevations in summer ($p < 0.05$, Figure 1). Taken together, it was clear that the availability of soil nutrients was significantly higher in summer than in winter, except for $NO_3^-$ (Figure 1).

*3.2. Microbial Biomass and Community Structure*

Soil microbial biomass carbon (MBC) at 700 m and 1200 m was significantly higher than that at 200 m, regardless of season ($p < 0.05$; Figure 2). We did not observe significant variation associated with elevation in soil microbial biomass nitrogen (MBN) in winter, whereas in summer, the MBN content was significantly lower at 200 m ($p < 0.05$; Figure 2). Overall, the soil MBC and MBN concentrations in summer were higher than those in winter. In winter, the soil MBC/MBN ratio was the lowest at the low elevation (200 m), and there was no significant difference between the elevations in summer.

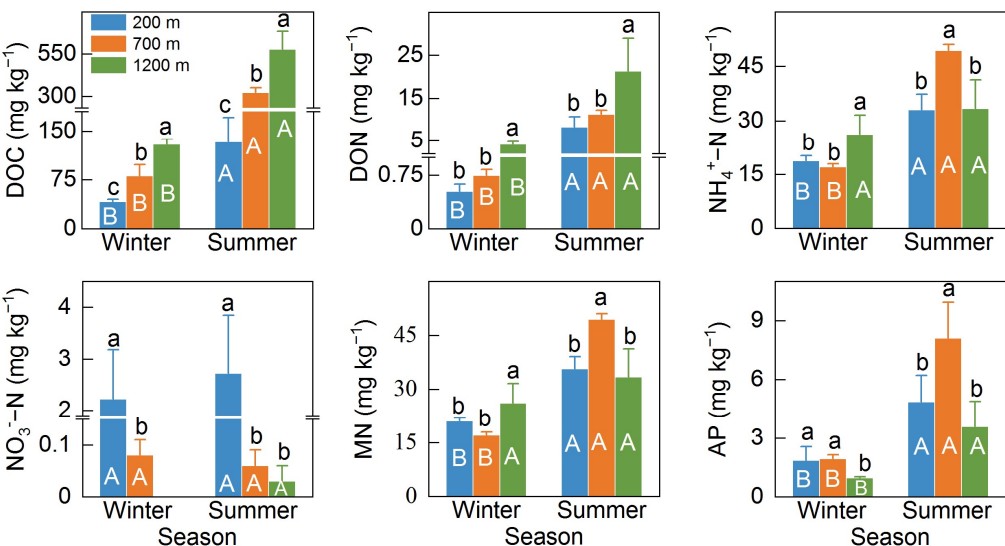

**Figure 1.** Soil carbon, nitrogen, and phosphorus availability in *Cunninghamia lanceolata* forests at different elevations during winter and summer. Values are means ± standard deviation (*n* = 4). DOC, dissolved organic carbon; DON, dissolved organic nitrogen; MN, mineral nitrogen; AP, available phosphorus. Different capital letters on the bar indicate significant differences between seasons at the same elevation, and different lowercase letters above the bar indicate significant differences between elevations during the same season (*p* < 0.05).

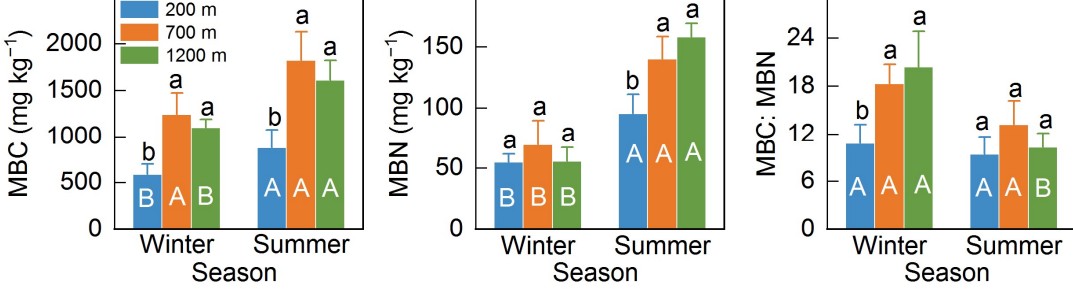

**Figure 2.** Soil microbial biomass carbon and microbial biomass nitrogen at different elevations. Values are means ± standard deviation (*n* = 4). Different capital letters on the bar indicate significant differences between seasons at the same elevation, and different lowercase letters above the bar indicate significant differences between elevations within the same season (*p* < 0.05). MBC, microbial biomass carbon; MBN, microbial biomass nitrogen; MBC:MBN: the ratio of microbial biomass carbon to microbial biomass nitrogen.

With the exception of fungi, the biomass of all of the PLFA biomarkers showed similar trends along the elevation gradient. The total PLFAs, Gram-positive bacteria (GP), Gram-negative bacteria (GN), and arbuscular mycorrhizal fungi (AMF) were higher at the medium elevation (700 m). The difference was significant in summer (*p* < 0.05; Figure 3), but we did not observe significant variation in winter. The fungi biomass showed an increasing trend with rising elevation both in summer and winter. Moreover, these biomarkers were higher in summer compared to in winter. With respect to the ratio of GP:GN, we observed that there were no distinctive differences between the elevation sites in summer, while it was the highest at 200 m in winter (*p* < 0.05; Figure 3). The ratio of F:B was the highest at 1200 m in both seasons, and seasonal differences only existed at this elevation. There was no significant interaction effect between elevation and season on the total PLFAs, GP, GN, AMF, ACT, fungi biomass, and F:B, with the exception of GP:GN (Figure 3).

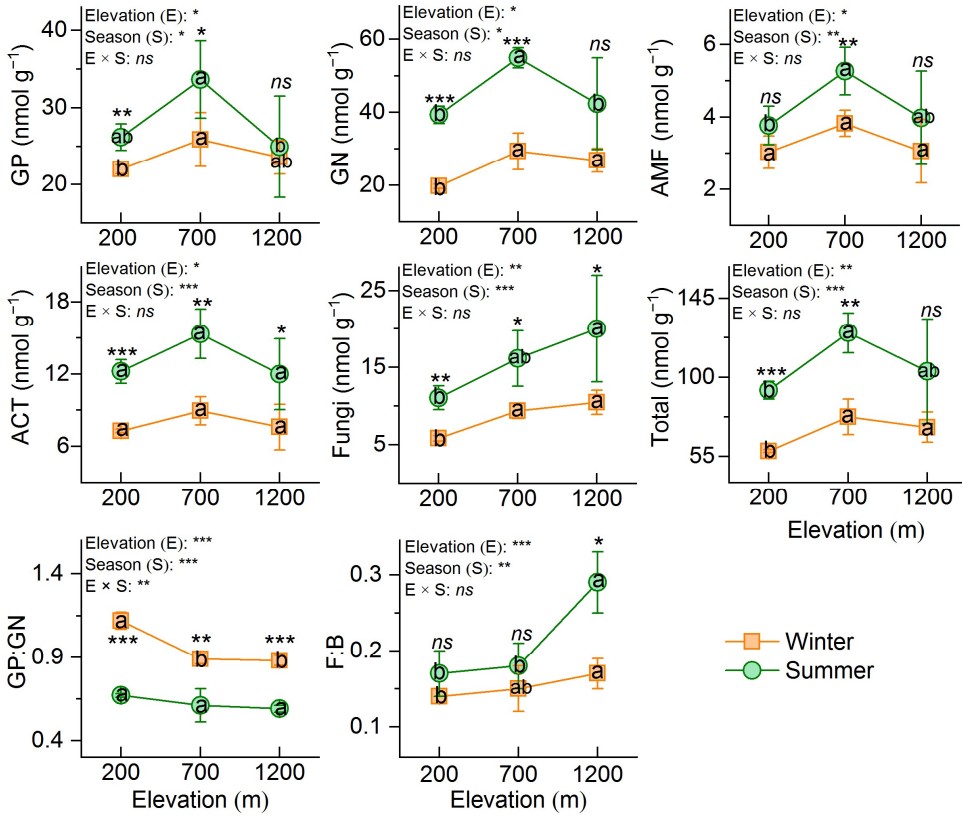

**Figure 3.** Effects of elevation and season on the abundance of phospholipid fatty acid biomarkers (in nmol g$^{-1}$ soil). Values are means ± standard deviation ($n = 4$). Different lowercase letters in the same season (winter or summer) represent significant differences ($p < 0.05$) among three elevations. * above the bar indicates significant differences between winter and summer at the same elevation; ns, no significant difference; *, $p < 0.05$; **, $p < 0.01$; ***, $p < 0.001$. GP, Gram-positive bacteria; GN, Gram-negative bacteria; AMF, arbuscular mycorrhizal fungi; ACT, actinomycetes; Total, total microbial PLFAs; GP:GN, the ratio of Gram-positive bacteria to Gram-negative bacteria; F:B, fungi to bacteria ratio.

### 3.3. CUE and Soil Enzyme Activity

The variation trend of soil CUE with elevation was opposite in winter and summer, and the interaction between season and elevation on CUE was significant ($p < 0.05$; Figure 4). C-degrading enzymes were significantly higher at 1200 m than at 700 m or 200 m, and there was no significant difference between winter and summer, except for at 700 m ($p < 0.05$; Figure 4). In winter, the N-degrading enzymes increased with increasing elevation, while in summer, they were the lowest at 700 m, and the seasonal differences were significant only at the high elevation ($p < 0.05$; Figure 4). In addition, there were interactive effects on the C-degrading enzymes and N-degrading enzymes between elevation and season ($p < 0.05$). For *p*-degrading enzyme activity, it was also the highest at 1200 m, and there were significant differences among different seasons ($p < 0.05$; Figure 4), but elevation and season had no interaction on it ($p > 0.05$).

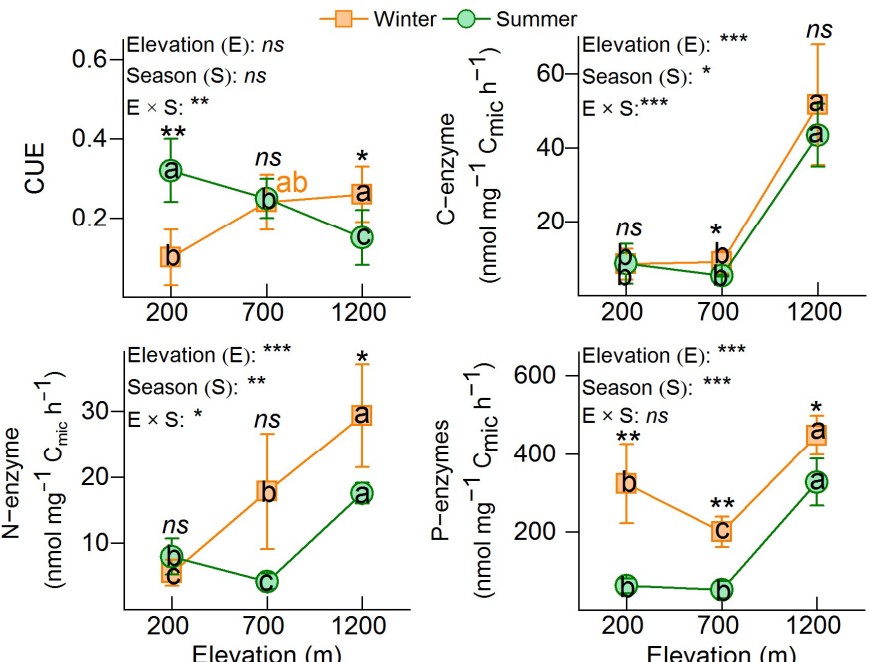

**Figure 4.** Soil microbial C use efficiency and specific potential activity of the hydrolytic enzymes (nmol mg$^{-1}$ Cmic h$^{-1}$) involved in C, N, and P acquisition at different elevations. The values are means $\pm$ standard deviation ($n = 4$). CUE, carbon use efficiency; Cmic, microbial biomass C; C-enzyme, including βG (β-1, 4-glucosidase) and CBH (cellobiohydrolase); N-enzyme, NAG (β-1, 4-N-acetylglucosaminidase); P-enzyme, AP (acid phosphatase). Different lowercase letters in the same season (winter or summer) represent significant differences ($p < 0.05$) among three elevations. * above the bar indicates significant differences between winter and summer at the same elevation; ns, no significant difference; *, $p < 0.05$; **, $p < 0.01$; ***, $p < 0.001$.

### 3.4. Factors Drive Seasonal Variation in Soil Microbial Community Composition

The correlations were plotted to explore the relationship of the PLFAs with different variables (Figure 5). There was no significant correlation between MBN, AMF, and soil properties in winter (Figure 5a). In addition to SOC, TN, and TP, soil fungi biomass was positively correlated with soil moisture ($p < 0.01$), DOC ($p < 0.001$) and DON ($p < 0.05$) and negatively correlated with soil temperature ($p < 0.05$). The ratio of GP:GN was significantly negatively correlated with soil moisture ($p < 0.05$) and DOC ($p < 0.01$); the F:B ratio was positively correlated with the concentration of soil moisture, DON ($p < 0.05$), and DOC ($p < 0.01$) and negatively correlated with soil temperature ($p < 0.01$) and soil pH ($p < 0.05$; Figure 5a). In summer, it was observed that the PLFA biomarkers were significantly positively correlated with available N and AP, except for fungi (Figure 5b). The fungi showed a negative relationship to the Ts and pH and were significantly positively correlated with the contents of DOC and DON as well as with the F:B ratios (Figure 5b). There was no relationship between the GP:GN ratios and the soil properties.

Redundancy analysis (RDA) showed that the composition of the soil microbial community in winter was related to soil DOC, explaining 35.5% of the variance in the composition ($p = 0.02$; Figure 6a and Table A1). The selected soil properties explained 66.54% of the variation in the microbial community composition in winter (Figure 6a). The composition of soil microbial community in summer was related to MN, which explained 48.9% of the variance ($p = 0.002$), and to pH and SM, which explained 24.3% ($p = 0.006$) and 12.4% ($p = 0.002$) of the variance, respectively. Together, these selected environmental variables explained 90.94% of the variations in the microbial community composition in summer (Figure 6b; Table A1).

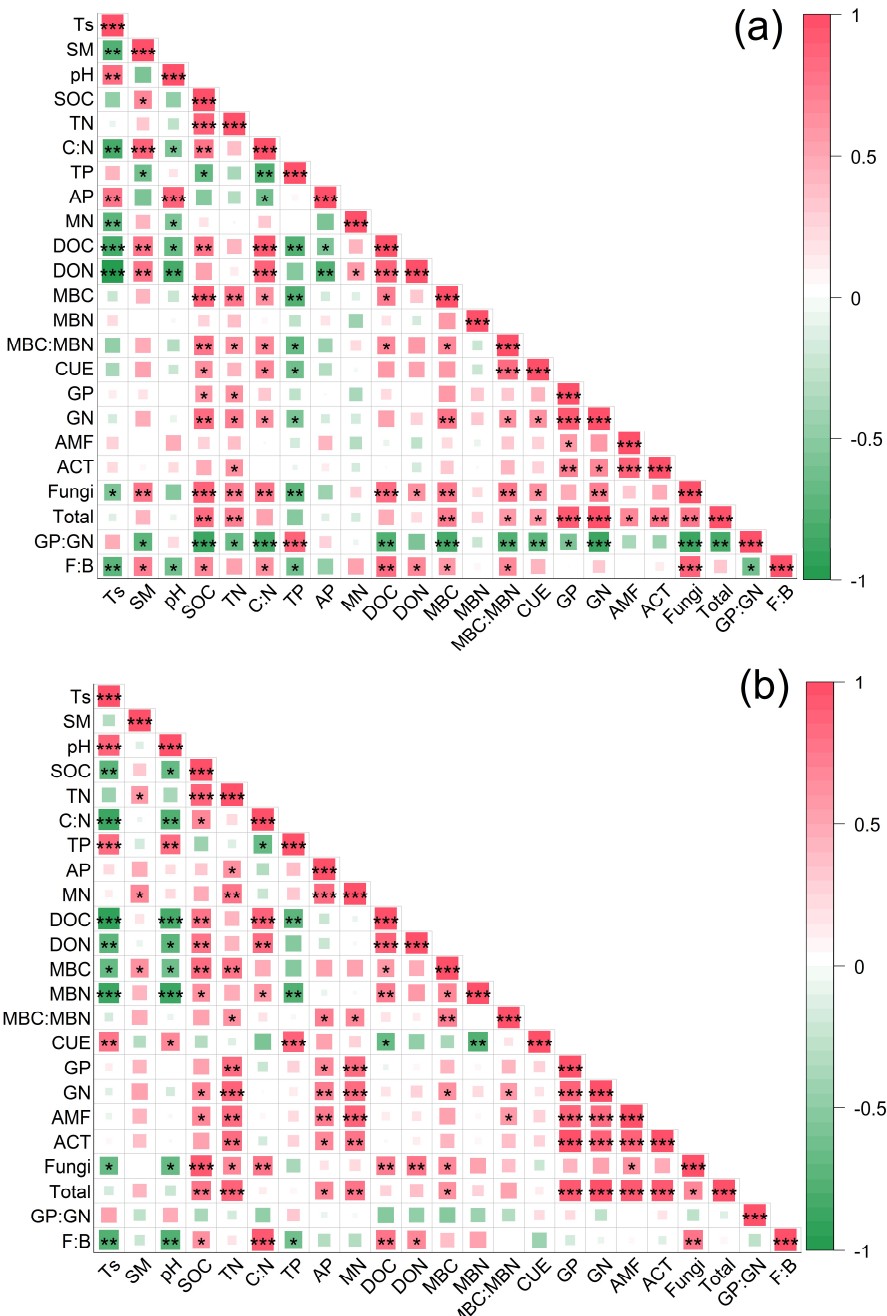

**Figure 5.** Correlation heat map between soil physicochemical properties and microbial index for (**a**) the winter and (**b**) summer. Red and green squares indicate positive and negative relationships, respectively. The figure shows the Pearson's correlation coefficients; * indicates $p < 0.05$, ** indicates $p < 0.01$, *** indicates $p < 0.001$. Ts, soil temperature; SM: soil moisture; SOC, soil organic carbon; TN, soil total nitrogen; TP, soil total phosphorus; AP, available phosphorus; MN, mineral nitrogen; DOC, dissolved organic carbon; DON, dissolved organic nitrogen; MBC, microbial biomass carbon; MBN, microbial biomass nitrogen; MBC:MBN: the ratio of microbial biomass carbon to microbial biomass nitrogen; CUE: carbon use efficiency; GP, Gram-positive bacteria; GN, Gram-negative bacteria; AMF, arbuscular mycorrhizal fungi; ACT, actinomycetes; Total, total microbial PLFAs; GP:GN, the ratio of Gram-positive bacteria to Gram-negative bacteria; F:B, fungi to bacteria ratio.

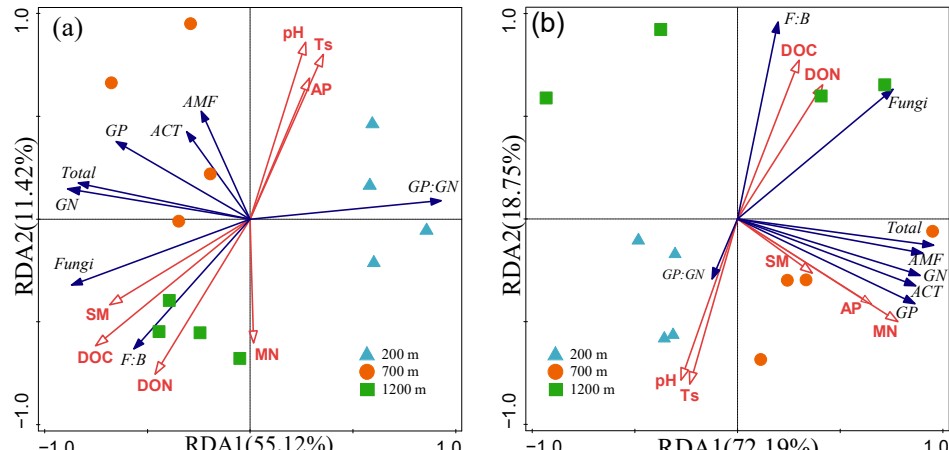

**Figure 6.** Redundancy analysis ordination biplot of PLFA profiles indicating the relationships between the variation in microbial community composition and environmental parameters: (**a**) winter and (**b**) summer. GP, Gram-positive bacteria; GN, Gram-negative bacteria; AMF, arbuscular mycorrhizal fungi; ACT, actinomycetes; Total, total microbial PLFAs; GP:GN, the ratio of Gram-positive bacteria to Gram-negative bacteria; F:B, fungi to bacteria ratio; Ts: soil temperature; SM: soil moisture; DOC, dissolved organic carbon; DON, dissolved organic nitrogen; MN, mineral nitrogen; AP, available phosphorus.

The SEM with Ts, DOC content, and SM on the elevation gradient could explain 71% of the total variance in the PLFAs in winter (Figure 7a). Elevation was negatively associated with Ts but positively correlated with DOC and SM. Ts and DOC showed a positive relationship with PLFAs, whereas Ts did not affect PLFAs. Structural equation modeling analysis of summer (Figure 7b) suggested elevation had a negative correlation with Ts, pH, and CUE and had no significant effect on MN. Elevation also indirectly affected PLFAs by affecting soil pH. Meanwhile, MN also had a significant positive effect on the PLFAs.

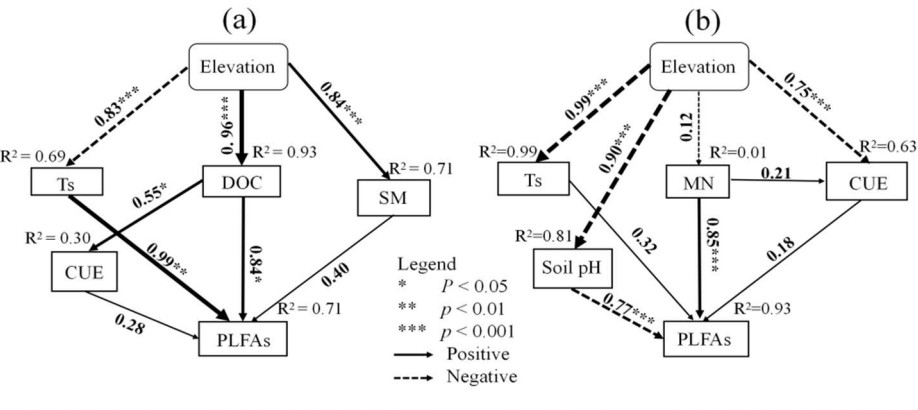

**Figure 7.** Structural equation model (SEM) was used to analyze the effects of environmental factors on soil microbial community along the elevation gradient: (**a**) winter and (**b**) summer. The solid and dotted arrows represent significant positive and negative pathways, respectively. Numbers at arrows are standardized path coefficients, and arrow width is proportional to the strength of the relationship. $R^2$ values on top of response variables indicate the proportion of variation explained by relationships with other variables. Ts, soil temperature; DOC, dissolved organic carbon; SM, soil moisture; MN, mineral nitrogen; CUE, carbon use efficiency.

## 4. Discussion

Soil microorganisms are the most active component of soil ecologies, and the number of soil microorganisms is mainly affected by climate, soil texture, physical and chemical properties, and vegetation types [45]. In this study, the soil MBC content at 700 m and 1200 m was shown to be significantly higher than that at 200 m, regardless of season. This may be related to the higher soil organic matter (SOM) at 700 and 1200 m than at 200 m. The turnover rate of SOM is crucial to the carbon cycling process, and it is generally believed that the turnover rate of SOC will accelerate with the increase in temperature [46]. Changes in climate and soil properties along the elevation gradient can also indirectly affect the formation and accumulation of SOC through microbial decomposition and turnover [28,47,48]. In our study, the low temperature conditions at the high elevation made the turnover time long, which is more conductive to the accumulation of SOC. Previous studies have shown that SOM is an important factor affecting soil microbial biomass [49], and a higher SOM can provide more optimal material and energy sources for microbial metabolism [2,50]. The higher the content of organic matter, the better material and energy sources provided for microbial metabolism [51]. Soil microbial biomass was closely related to the eco-environmental factors of the habitat. Kemmit et al. [52] reported that the soil pH decreased with the increase in elevation, and the soil acidity gradually increased, resulting in the reduction of the mineralization rate of hydrolyzed nitrogen, which negatively affected the growth and reproduction of soil microorganisms. However, in our study, there was a significant negative correlation between soil MBN and pH in summer (Figure 6b). In addition, we observed that the soil MBC:MBN ratio increased with elevation in winter. A deficiency of P has been reported to increase the MBC:MBN ratio [53]. This can be supported by our finding of a significant and negative correlation between the soil MBC:MBN and soil TP content (Figure 6a). Furthermore, high MBC:MBN ratios often indicate poor P availability in the presence of high C availability [54], but other nutrient deficiencies may also lead to high MBC:MBN ratios [55,56]. From the perspective of season, soil MBC and MBN were significantly higher in summer than in winter (Figure 3). Compared to winter, higher soil temperature and humidity in summer provided a good metabolic environment for the reproduction of soil microorganisms, thus increasing soil MBC and MBN.

Elevation influences the functional diversity and metabolic activity of soil microbial communities by improving soil physicochemical properties and nutrient cycling processes [57,58]. Consistent with our hypotheses, the variation trend of soil microbial community composition along the elevation gradient was not completely consistent in winter and summer. The distribution pattern of soil microorganisms in *Cunninghamia lanceolata* forest at different elevations showed that the GN biomass and total PLFAs at the low elevation in winter were significantly lower than other elevations and that the soil fungal community increased with elevation (Figure 3). In forest soils, fungi can decompose more complex organic matter than soil bacteria [59,60], and the C required to form fungal biomass is more than bacteria [61], so substrates with high C:N ratio are more conducive for fungal growth [33,62]. The soil C:N ratio increased with elevation, and the correlation analysis showed that the soil fungal community had a significant positive correlation with the soil C:N ratio (Figure 5a). In addition, studies have found that fungi tend to have higher CUE than bacteria [63,64]. The growth of soil microorganisms depends directly on microbial CUE [65]. In winter, the soil microbial CUE increased with increasing elevation (Figure 4), which indicates that microbes may utilize much more C as their energy requirement to face the low temperatures at high-elevation sites than at low-elevation sites. At the same time, the redundancy analysis results showed that soil DOC is the main factor affecting soil microbial community at different altitudes in winter (Figure 6a). As the most important energy source for the growth of soil microorganisms, soil DOC can be directly absorbed and utilized by microorganisms, while litter and root C-input (i.e., exudates) are the import C sources of soil microbes [66–68]. In our study, the soil DOC content increased significantly with altitude, while in winter, plant growth was slow, and the DOC content imported by

root exudates was low. At the same time, the low temperature and low water content of the soil reduces the effective carbon source and the mobility of nutrients. Compared to the low elevation, the higher DOC content at the medium and high elevations provided more energy support for microbial growth. Previous studies showed that in soil with low nutrient availability, GP bacteria forming oligotrophic communities preferentially used the recalcitrant C component, while GN bacteria were more popular in the soil with a high nutrient content [69–72]. We believe that this may be the main reason why the GP:GN ratio at the low elevation was significantly higher than that at the medium and high elevations in winter.

In our study, we found that bacteria, AMF, and total PLFAs were significantly higher at the medium elevation in summer (Figure 3). The results of the redundancy analysis showed that soil N availability was an important factor affecting the composition of the microbial community and had significant positive effects on all of the soil microbial communities, except for fungi (Figure 6b). Soil-soluble nutrients are more readily available for uptake by microorganisms and are sensitive to seasonal changes [73,74]. The high temperature and rainy climate conditions in the subtropical region in summer accelerate the decomposition and leaching of litter, thus promoting an increase in soil carbon, especially in the active carbon source [67,68]. Summer is the peak period of plant growth, and the large demands of plants for soil nutrients limit the availability of soil nutrients by soil microorganisms, forming a competitive relationship [75]. Due to the high dependence of GP and GN bacteria on soil inorganic nitrogen, they speed up the growing rate, increase microbial biomass, and alter community proportions at medium elevations, where available nitrogen levels are high [76]. In addition, we observed a similar response of soil fungal biomass to changes in elevation as in winter (i.e., an increase with elevation; Figure 5b), which was probably affected by the soil's pH value. A lower pH can increase the solubility of soil organic matter and change the composition of dissolved organic matter [77], which could affect the soil fungal community. Many studies indicate that the change in soil pH drives the dominant bacterial and fungal communities [78,79]. Generally, bacterial groups are suitable for growth under slightly alkaline conditions, while fungi are more resistant to acid soil environment [80,81], which was consistent with our result that soil pH was a significant factor influencing microbial community structure in summer according to the RDA data. The pH value tends to be lower at high elevations, which leads to an increase in soil F:B. Generally, we found that the impact of the input source of carbon substrates on the proportion of microbial groups is more influential in winter than in summer.

## 5. Conclusions

Using an elevation gradient with similar annual precipitation but a clear temperature gradient, we demonstrated how rising temperatures affect the soil microbial communities in *Cunninghamia lanceolata* forests. The results indicate that most of the edaphic properties vary with seasonal changes and influence the composition and structure of soil microbial communities along the elevation gradient. The variation trend of the soil microbial community along the elevation gradient varied between winter and summer. In winter, the Gram-negative bacteria and total PLFAs were the lowest at 200 m and fungi biomass increased with increasing elevation, with soil DOC being the main factor controlling microbial community composition. In summer, with the exception of ACT and fungi, the biomass of other microbial communities was the highest at the medium elevation, with soil N availability being the most important factor driving the change in the microbial community across the elevation gradient. Moreover, in summer, microbial carbon use efficiency (CUE) increased with increasing elevation, whereas an opposite trend was observed in winter. Taken together, our results suggest that N and C availability drive the seasonal variation in soil microbial community composition across a subtropical elevation gradient. Our findings will help us to better understand the role of subtropical forest microbial communities in providing ecosystem services and the interactions between plants and microbes under the background of global climate change.

**Author Contributions:** X.X., M.L. and J.X. wrote the manuscript; M.L. and J.X. designed and supervised the experiment; X.X., C.D., X.L., Y.L., W.L. and Y.J. carried out the measurements following the field incubation and collected and analyzed the data. All authors have read and agreed to the published version of the manuscript.

**Funding:** This research was funded by the National Natural Science Foundation of China (Nos. 32001169, 31870604, and 32030073).

**Institutional Review Board Statement:** Not applicable.

**Informed Consent Statement:** Not applicable.

**Data Availability Statement:** The data that support the findings of this study are available in the Appendix A of this article.

**Acknowledgments:** We thank Yangyi Nie, Yinbang Ren, Zuoqing Fu, Chaoyue Ruan, and Qiaoming Huang for helping with the laboratory and field work.

**Conflicts of Interest:** The authors declare that they have no competing interest.

## Appendix A

**Table A1.** Correlations between environmental parameters and RDA ordination of PLFAs at different elevations. Significant differences are indicated by bold-face $p$ values, and italics indicate marginal significance.

| Environment Parameters | Explains % | Pseudo-F | $p$ |
|---|---|---|---|
| Winter | | | |
|     DOC | 35.5 | 5.5 | **0.020** |
|     Ts | 15.9 | 3.0 | *0.096* |
|     SM | 8.6 | 1.7 | 0.218 |
| Summer | | | |
|     MN | 48.9 | 9.6 | **0.002** |
|     pH | 24.3 | 8.2 | **0.006** |
|     SM | 12.4 | 6.9 | **0.002** |
|     Ts | 3.7 | 2.5 | 0.100 |
| DON | 2.4 | 1.7 | 0.176 |

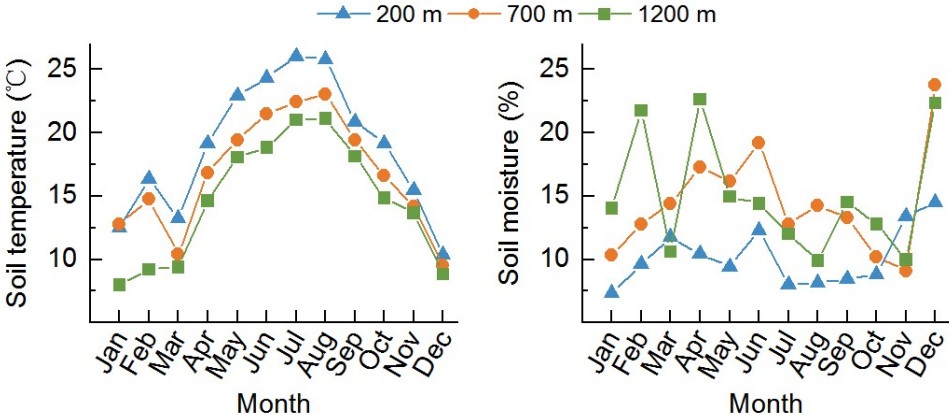

**Figure A1.** Monthly soil temperature and moisture at different elevations.

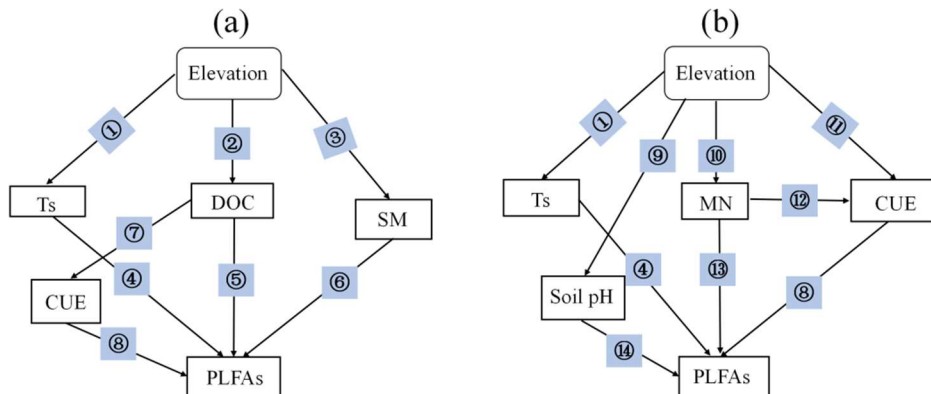

**Figure A2.** A priori model showing the rationale behind the direct and indirect associations from elevation, soil temperature and moisture, soil pH, C and N availability, and CUE with PLFAs: (**a**) winter and (**b**) summer.

**Table A2.** Basic rationale and related references for prior models.

| # | Rationale | References |
|---|---|---|
| 1, 3 | Changes in elevation affect soil temperature and moisture. | [16,82] |
| 2, 10 | Soil nutrient availability and spatial heterogeneity were significantly affected by elevation. | [14] |
| 4 | Warming will change the composition of the soil microbial community. | [83,84] |
| 5, 13 | Soil DOC is an important carbon source for microbial growth. Soil N availability affects the microbial community structure. | [66,85] |
| 6 | Soil moisture is the major factor influencing microbial community structure. | [86] |
| 7, 12 | Soil microbial CUE will increase with nutrient availability. | [87] |
| 8 | The growth of soil microorganisms directly depends on microbial CUE. | [65] |
| 9 | Elevation has a significant effect on soil pH. | [81] |
| 11 | Soil microbial carbon use efficiency was significantly different at different altitudes. | [88] |
| 14 | It is well-known that soil pH is an important factor affecting microbial community composition. | [79] |

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
