# Peer review of "Carbon and Nitrogen Availability Drives Seasonal Variation in Soil Microbial Communities along an Elevation Gradient"

_forests, doi:10.3390/f13101657_

Round 1
Reviewer 1 Report
This manuscript aims at examining seasonal resource availability for heterotrophic microbial communities in soil with increasing elevation. This study demonstrates distinct seasonal changes in the soil microbial community composition across an elevation gradient which are driven by carbon and nitrogen resources availability and shifting in microbial CUE. Further I think the research topic will be interesting for the readers of the Journal. The paper is well written and easy to understand, and the statistics used are appropriate.
Statistical Analysis
1. Software of performing sem models should be added
2. A priori models for the structural equation modeling analysis should be added in the supplementary materials.
Results
The words and letters in the Fig.5 cross, which affects the reading of the figure and needs to be adjusted.
Discussion
The discussion generally needs improvement. Altitude also affects geochemical properties and, in turn, organic carbon turnover and microbial activity. e.g. Climate and geochemistry interactions at different altitudes influence soil organic carbon turnover times in alpine grasslands
Author Response
Response to Review Comments
Reviewer # 1
The authors would like to thank you for your valuable suggestions. The manuscript has been revised in light of your comments. The writing has been substantially improved to meet your publication quality standards.
Comments and Suggestions for Authors
This manuscript aims at examining seasonal resource availability for heterotrophic microbial communities in soil with increasing elevation. This study demonstrates distinct seasonal changes in the soil microbial community composition across an elevation gradient which are driven by carbon and nitrogen resources availability and shifting in microbial CUE. Further I think the research topic will be interesting for the readers of the Journal. The paper is well written and easy to understand, and the statistics used are appropriate.
Response: We appreciate the overall positive comment about our manuscript. At the same time, as for your valuable suggestions, we actively adopt and carefully modify them.
Statistical Analysis
- Software of performing sem models should be added.
Response: The software used in the structural equation model has been described in the Statistical
Analysis. We used a structural equation modeling (SEM) with AMOS 24.0 (AMOS Development Corporation,
Chicago, IL, USA) to examine the key factors driving seasonal variations in microbial community structure.
- A priori models for the structural equation modeling analysis should be added in the supplementary materials.
Response: The prior models for structural equation models have been added in the supplementary
materials (Figure. A2)
Results
The words and letters in the Fig.5 cross, which affects the reading of the figure and needs to be adjusted.
Response: The position of the words in Fig. 5 has been adjusted.
Discussion
The discussion generally needs improvement. Altitude also affects geochemical properties and, in turn, organic carbon turnover and microbial activity. e.g. Climate and geochemistry interactions at different altitudes influence soil organic carbon turnover times in alpine grasslands
Response: Thank you for your valuable comments. In terms of discussion, there is indeed a lack of discussion related to soil organic carbon. As you said, changes in altitude not only cause changes in climate, but also have an impact on soil properties, etc., which in turn affects the turnover of soil organic carbon. As the largest carbon pool in terrestrial ecosystems, both the input and output of soil carbon pools affect the carbon cycle processes. At the same time, higher organic matter can provide more optimal material and energy sources for microbial metabolism. Therefore, we have additions in the manuscript regarding the lack of this aspect.
Reviewer 2 Report
This was a huge work, but some figures are not well prepared.
The GC parameters are missing (PLFA determination) please put it.
Which Spectrophotometer was used for the other determination?
I recommend to all figures to rework because the manuscript in black and white is hard to see the differences in the figures.
Figure 5 is not so clear I recommend improving it. (Especially in black and white) I recommend a better position of the column names. (above of the column)
Use “mol L-1” instead of M
Author Response
Response to Review Comments
Reviewer # 2
The authors would like to thank you for your valuable comments. The manuscript has been revised in light of your suggestions. I hope that the revised manuscript will meet with your approval for publication in Forests.
Comments and Suggestions for Authors
This was a huge work, but some figures are not well prepared.
Response: At first, we appreciate the overall positive comments on our manuscript. In addition, we have improved some deficiencies according to your suggestions.
The GC parameters are missing (PLFA determination) please put it.
Response: As you mentioned, we have made corresponding supplements to
the GC parameters related to PLFA determination.
Which Spectrophotometer was used for the other determination?
Response: The spectrophotometer used in the determination of related indexes has been supplemented in the manuscript.
I recommend to all figures to rework because the manuscript in black and white is hard to see the differences in the figures.
Response: We have modified the image format to make it easier for you to read.
Figure 5 is not so clear I recommend improving it. (Especially in black and white) I recommend a better position of the column names. (above of the column).
Response: Figure 5 has been modified based on your comments.
Use “mol L-1” instead of M
Response: Corrected accordingly.